# A Benchmark Dataset for Evaluating Practical Performance of Model Quality Assessment of Homology Models

**DOI:** 10.3390/bioengineering9030118

**Published:** 2022-03-15

**Authors:** Yuma Takei, Takashi Ishida

**Affiliations:** 1Department of Computer Science, School of Computing, Tokyo Institute of Technology, Ookayama, Meguro-ku, Tokyo 152-8550, Japan; takei@cb.cs.titech.ac.jp; 2AIST-Tokyo Tech Real World Big-Data Computation Open Innovation Laboratory (RWBC-OIL), National Institute of Advanced Industrial Science and Technology (AIST), Aomi, Koto-ku, Tokyo 135-0064, Japan

**Keywords:** model quality assessment, evaluation of model accuracy, protein structure prediction, machine learning, deep learning, MQA, EMA

## Abstract

Protein structure prediction is an important issue in structural bioinformatics. In this process, model quality assessment (MQA), which estimates the accuracy of the predicted structure, is also practically important. Currently, the most commonly used dataset to evaluate the performance of MQA is the critical assessment of the protein structure prediction (CASP) dataset. However, the CASP dataset does not contain enough targets with high-quality models, and thus cannot sufficiently evaluate the MQA performance in practical use. Additionally, most application studies employ homology modeling because of its reliability. However, the CASP dataset includes models generated by de novo methods, which may lead to the mis-estimation of MQA performance. In this study, we created new benchmark datasets, named a homology models dataset for model quality assessment (HMDM), that contain targets with high-quality models derived using homology modeling. We then benchmarked the performance of the MQA methods using the new datasets and compared their performance to that of the classical selection based on the sequence identity of the template proteins. The results showed that model selection by the latest MQA methods using deep learning is better than selection by template sequence identity and classical statistical potentials. Using HMDM, it is possible to verify the MQA performance for high-accuracy homology models.

## 1. Introduction

Protein structure prediction, which predicts the three-dimensional structure of a protein from its amino acid sequence, is important in structural bioinformatics. Various prediction methods have been developed to date [1,2], two major types of which are homology modeling and de novo modeling. Homology modeling predicts structures using the three-dimensional structure of a template protein, of which the amino acid sequence is similar to the target sequence. In contrast, de novo modeling predicts structures without using a template. While homology modeling cannot make predictions without the presence of a template structure, de novo modeling can make predictions regardless of the presence of a template. Furthermore, homology modeling is generally accurate when a good template exists [3,4], and the computational cost is much lower than that of de novo modeling. Therefore, homology modeling is often used in practical applications, such as drug discovery [5,6,7,8].

There are many methods for predicting protein structures. Each method is capable of generating multiple predicted structures in many cases. Therefore, it is necessary to select a structure from several structures for use in subsequent applications. Additionally, the user needs to judge whether the structure model has sufficient quality. However, in practical situations, these are not possible, because the ground truth structure has not been determined, and the accuracy of the predicted structure cannot be calculated.

To solve this problem, there is research called the model quality assessment (MQA). MQA estimates the accuracy of the predicted structures, enabling the comparison of structures and the verification of prediction accuracy. Relative accuracy is generally sufficient to select the most accurate predicted structure from multiple structures. However, it is also important to estimate the absolute accuracy of a single predicted model structure to determine whether it can be used in subsequent applications. Various MQA methods have been developed to date, and the recently developed methods often use deep learning [9,10,11]. Deep learning-based methods often show better accuracy than conventional methods, such as potential energy function-based methods. However, such learning methods tend to overfit the training datasets. Thus, if the training and test sets used are biased, the performance evaluation might lead to overestimation.

Some datasets are available for training and evaluating the performance of MQA methods, such as CAMEO [12], 3DRobot [13], and QUARK [14,15]. The most commonly used dataset among them is the critical assessment of protein structure prediction (CASP) [16] dataset. CASP is a benchmark for protein structure prediction and is revised every 2 years. Notably, CASP includes a structure prediction category and an MQA category. The three-dimensional structure models predicted in the structure prediction category are used in the MQA category. The data used in the MQA category are often used as benchmark datasets for MQA methods.

The CASP dataset is often used as a benchmark dataset in MQA research [9,10,11] because it contains several structure models for various target proteins, and researchers can directly compare their results with previous CASP experiments. However, there are some problems with this dataset. First, there are not enough targets that contain highly accurate model structures. In the CASP11-13 datasets [16,17,18], which are often used as a test dataset for MQA methods, 87 of the 239 targets had predicted model structures with global distance test total score (GDT_TS) [19] greater than 0.7, which is regarded as highly accurate. Only 19 targets had GDT_TS greater than 0.9, which is close to the experimental accuracy. A key capability of MQA methods in practical situations is the ability to select the most accurate model from the highly accurate models, such as those with GDT_TS greater than 0.7. However, the CASP dataset does not contain accurate models for more than half of the targets. Thus, it is not possible to fully evaluate the ability to select a sufficiently accurate model among several accurate models. Second, there are multiple structural models predicted by various protein structure prediction methods. For a single target in the CASP dataset, there are structural models predicted by approximately 30 different prediction methods, and each method has different characteristics. One problem with the inclusion of model structures from various prediction methods is that it is unclear whether the MQA method assesses the quality of the model structure or merely captures the characteristics of the prediction method. For example, it is possible that MQA methods that use Rosetta [20] energy as input features may overestimate the structure predicted by methods that are optimized for Rosetta energy. Thus, it is not clear whether the MQA method judges the quality of the prediction structure itself or the features of the prediction method when models are predicted by multiple methods. Third, the CASP dataset consists of structural models predicted by both modeling methods. Structure prediction methods are often used in drug discovery [21,22,23,24,25]. However, homology modeling has mainly been used because of its reliability. As the accuracy of de novo modeling has improved in recent years, the structures predicted by de novo modeling may be used for drug discovery in the future. However, de novo models are rarely used. Furthermore, it is important to evaluate the MQA performance of structures predicted by homology modeling. Importantly, the CASP dataset contains many structures predicted by de novo modeling. There are also decoy sets that consist only of structures predicted by homology modeling [26,27,28]. Unfortunately, these datasets have problems, such as a small number of targets, few structure models for each target, and a limited number of high-quality structure models. Additionally, while the CASP datasets include both single-domain and multi-domain proteins, most models are for single-domain proteins. The number of multi-domain proteins is insufficient.

The CAMEO dataset has also been often used as an evaluation dataset in recent MQA studies. The CAMEO dataset has more frequent updates and a larger number of targets than the CASP dataset. In addition, CAMEO’s Model Quality Estimation category contains 1280 predicted structures with global local distance difference test (lDDT) score [29] greater than 0.8 out of 6690 structures in one year (19 February 2021–12 February 2022), which means that CAMEO has more structures with higher accuracy than CASP. However, CAMEO has the problem that the number of predicted structures per target is small. The number of predicted structures per target in CAMEO is about 10, and the performance of selecting the best structure from among the structures for a single target cannot be fully evaluated.

In this study, we constructed a dataset named homology models dataset for model quality assessment (HMDM) for benchmarking MQA methods in practical situations. We used a single homology modeling method for tertiary structure prediction. The protein targets were selected to include the most accurate models. We then created two datasets—one containing single-domain proteins and another containing multi-domain proteins. After constructing the datasets, we compared the performance of the existing MQA methods using our datasets and determined their performance for highly accurate homology models.

## 2. Methods

We created a single-domain dataset and a multi-domain dataset to evaluate the MQA performance for single-domain and multi-domain proteins. We designed these datasets to contain a large number of high-quality models to evaluate the MQA performance in practical scenarios. To generate the high-quality models, we used a homology modeling method to predict the structure and selected target proteins with rich template structures. Then, to ensure an unbiased distribution of model quality for each target, structures were modeled using various templates, which were sampled to create the final dataset. Once the datasets were completed, we compared the MQA performance of the datasets using various MQA methods and indices of alignment quality, such as identity, between the target and template sequence.

The workflow for creating the datasets is shown in Figure 1. First, we selected template-rich entries as targets from the structural classification of proteins (SCOP) [30] and PISCES [31] databases, respectively. Then, the template was searched against protein data bank (PDB) [32] and modeled. Next, sampling was performed to ensure that the distribution of the model quality was not biased. Low-quality models were excluded. Finally, each target was confirmed to meet the criteria described later, and targets that did not meet the criteria were re-selected.

### 2.1. Dataset Construction

#### 2.1.1. Target Selection

We selected 100 targets from the SCOP version 2 (SCOP2) (released on 30 March 2021) database for the single-domain dataset and from the subset of PISCES server (released on 25 February 2021) for the multi-domain dataset to avoid redundancy among the targets.

SCOP2 classifies protein domains based on their evolutionary and structural relationships. Because SCOP2 has entries for each protein domain, we used it as the target selection source for the single-domain dataset. We selected one target from each protein superfamily to avoid target redundancy. There are four protein types in SCOP: globular, fibrous, membrane, and intrinsically disordered. We selected only globular proteins as targets, because fibrous and membrane proteins are not stable in their own protein structure domain, and intrinsically disordered proteins are inappropriate targets for structure prediction. Furthermore, SCOP classifies entries into five classes based on their secondary structure: all alpha, all beta, alpha/beta, alpha+beta, and small proteins. We excluded small proteins because of few entries. Then, we selected 25 targets equally from the other four classes, with 100 targets in total. When choosing a superfamily for each class, we chose superfamilies with a high number of entries. When selecting targets from the superfamily entries, we selected an entry with the highest number of hits in homology searching using three iterations of PSI-BLAST [33] v2.9.0.

PISCES is a server that can extract subsets of proteins using sequence identity and structural quality criteria. We used the subset which was precompiled using the following parameters: identity less than 20%, resolution less than 2.0, and R-factor less than 0.25. Since entries in PISCES are listed by amino acid sequence and contain both single-domain and multi-domain proteins, we extracted only multi-domain proteins based on the CATH [34] classification. To select template-rich entries as targets, we performed a homology search on each multi-domain entry in the same way as for the single-domain dataset, and selected 100 targets in the order of the number of hits.

#### 2.1.2. Homology Modeling

We used MODELLER [35], which is a commonly used homology modeling method, as a structure prediction method. Although there are several homology modeling methods, we chose a single method to evaluate whether the MQA methods capture the quality of the model, rather than the characteristics of the predicted structure for each structure prediction method.

We performed three iterations of PSI-BLAST against PDB (released on 7 April 2021) and selected template structures from each iteration. We used two methods to select template structures from the results of each iteration of PSI-BLAST. One method was to simply select the hits in the order of their e-values to select the template structure that is most similar to the target sequence. The other method was to cluster the hit sequences using CD-HIT [36,37] with a 95% threshold and select hits in the order of their e-values so that the clusters do not overlap, which allowed us to select various template structures. Up to 10 templates were selected from each iteration by each of these two methods, totaling up to 60 templates. Note that the templates were selected in order starting from the first iteration, and the hits selected in the previous iteration were not re-selected. Even if the template structure was the same, if the sequence alignment changed, the template was selected.

We excluded some hits when selecting the templates. First, proteins with the same PDB ID as the target proteins were excluded. Next, hits with less than 60% coverage were excluded because high-quality model structures were not generated from such templates. Finally, hits with more than 95% identity to the target sequence were excluded because their structures could have been determined for the exact same protein or they may have the same structure with only a few residue differences.

After selecting the templates, we generated five models with different random seeds of optimization for each template. At that time, we set automodel.md_level to the default (refine.very_fast) for model refinement. Thus, up to 300 models were generated for a single target.

#### 2.1.3. Sampling

After modeling, there was bias in the quality distribution of the model structures. Therefore, we sampled the models to reduce the bias. Before sampling, models with GDT_TS less than 0.4 were removed because their prediction quality was low and they were not suitable for evaluating the MQA performance.

Sampling was performed by limiting the number of models around the best model that had the best GDT_TS. If there were many models close to the best model, it was easy to select the model that is close to the best, and it was not possible to accurately evaluate the MQA performance when selecting the best model. Specifically, models whose GDT_TS difference from the best model was within 0.03 were defined as the models around the best. Up to 10 of these models were randomly sampled. Note that the best model was always included in the dataset separately from the models around the best model. The other models were randomly sampled so that the maximum number of models was 150, including the models already selected in the above procedure.

#### 2.1.4. Verification

Finally, a subset of each target was evaluated to determine whether it met the following criteria. Targets that did not meet the criteria were re-selected because they were not suitable for the purpose of this study. First, targets with fewer than 50 models after sampling were replaced for both single-domain and multi-domain datasets. Then, targets whose GDT_TS of the best model was less than 0.7 were replaced. When replacing a single-domain target, the entry in the same superfamily that had the next highest number of hits in the PSI-BLAST template search was selected as a target. When replacing multi-domain targets, we selected the entry with the next highest number of hits among the multi-domain entries for a target. We iteratively replaced targets until all targets met the criteria.

### 2.2. MQA Performance Evaluation for the Constructed Datasets

We compared the MQA performance of several MQA methods and template quality metrics, such as identity, in the datasets created using the procedure described above.

#### 2.2.1. Evaluation Metrics

We chose metrics that are important in practical situations as the main evaluation metrics. Some metrics that are important in practical situations included the ability to select the best model from a set of models and how accurately we could predict the value of GDT_TS. Therefore, we used the average of GDT_TS loss and the average of mean absolute error (MAE) per target as the main evaluation metrics. GDT_TS loss is the difference between the GDT_TS of the best model and the model with the highest MQA score. The lower the GDT_TS loss, the model that is closest to the best model is selected. If there were multiple highest MQA scores, we averaged their losses. The MAE is the average of the errors between the GDT_TS and MQA scores for each model. We also used the average Pearson and Spearman correlations between the GDT_TS and MQA scores for each target. GDT_TS was calculated using the TM-score [38]. To confirm statistical significance, Wilcoxon signed-rank tests were conducted using a significance level of 0.01.

#### 2.2.2. Evaluation Methods

We compared three method types: indices of the alignment quality between target and template sequence, statistical potential function-based MQA methods, and machine learning-based MQA methods. All of these methods were run in our local environment.

We used three indices of the alignment quality between the target and template sequences: identity, positive, and coverage. Identity generally indicates the percentage of residues that match within the aligned sequence, but in this study, identity was calculated by dividing the number of residues that match between the template and target sequences by the length of the target sequence. Positive indicates the percentage of residues with a positive alignment score, which is also generally dependent on the length of the alignment. In this study, we calculated the positive percentage using the length of the target sequence. We calculated the coverage as the ratio of the template sequence coverage to the length of the target sequence.

For statistical potential function-based MQA methods, we used discrete optimized protein energy (DOPE) [39] and statistically optimized atomic potentials (SOAP) [40]. DOPE is a method based on the distance potential between atoms and is used as the internal score function in MODELLER. SOAP uses the atomic distance and orientation between a pair of covalent bonds.

As machine learning-based methods, we selected ProQ3D [41], SBROD [42], P3CMQA [43], and DeepAccNet [44]. ProQ3D is a standard MQA method and is a neural network-based method that uses sequence profiles and Rosetta energy as inputs. Several versions were trained with different labels. We used the S-score version. SBROD is a ridge regression-based method that uses the structural features of the main chain as the input. P3CMQA is a three-dimensional convolutional neural network (3DCNN)-based method that uses atom-type features and sequence profiles as inputs. DeepAccNet is a method composed of 3DCNN and 2DCNN that uses distance-based and sequence-based features as inputs. Two versions of DeepAccNet were used: DeepAccNet, which is the standard version, and DeepAccNet-Bert, which uses bert-embeddings by ProtTrans [45]. ProQ3D and SBROD were selected because of their good performance in CASP13 [46]. P3CMQA and DeepAccNet were selected because they had good results in CASP14 [47].

## 3. Results

First, we described details on the constructed datasets. Then, we compared the prediction performance of the MQA methods on the datasets.

### 3.1. Constructed Datasets

We constructed single-domain and multi-domain datasets containing 100 targets each. The list of targets for the single-domain and multi-domain datasets are shown in Appendix A, respectively. Figure 2 shows the maximum GDT_TS value distribution for each target. For the both datasets, we generated structural models with GDT_TS greater than 0.9 for more than half of the targets. Compared to the CASP11-13 [16,17,18] dataset, both constructed datasets contain more accurate prediction structures (see Appendix A). More detailed information, such as the distribution of GDT_TS of models for each target, is available in the Appendix A.

### 3.2. Correlation between GDT_TS and the Alignment Quality

In homology modeling, the alignment quality between the target and template sequences is an important factor that affects the model quality. Therefore, the template is often selected based on the alignment quality. In this study, we used various template structures to model a single target and examined the relationship between the alignment and model quality. Figure 3 shows a scatter plot of GDT_TS and the three indicators of alignment quality: identity, positive, and coverage. There was a positive correlation between the alignment quality and GDT_TS. However, even if the identity and positive values were high, GDT_TS was low in some cases. In contrast, there were cases where the GDT_TS was high, even if the identity was low. Therefore, it was difficult to judge the quality of the predicted model only on the alignment quality.

### 3.3. MQA Performance Evaluation for the Constructed Datasets

Previous MQA studies generally evaluated their performance using the CASP dataset, and their performance for high-quality homology models was unclear. Thus, we benchmarked the performance of the existing MQA methods. Additionally, most practical studies using homology modeling often used sequence identity or the e-value to select the best template and structure model. We therefore compared the selection performance of alignment quality indices with those of the MQA methods of the constructed datasets.

The results for the single-domain dataset are listed in Table 1. Among the three indices of alignment quality (identity, positive, and coverage), the performance of identity was the best in terms of loss. The performance of the positive was better in terms of the Pearson and Spearman correlations. The performance of the coverage was lower than those of the other indices. Statistical potential-based methods (DOPE and SOAP) performed slightly better than identity for all indices. Classical machine learning-based methods (ProQ3D and SBROD) performed worse to loss. This is because structures with high identity are often accurate, which makes this is a good index for selecting the best model. In contrast, structures with low identity were not always inaccurate and had decreased performance for the Pearson and Spearman correlations. Newer deep learning-based methods (P3CMQA and DeepAccNet) performed better than alignment quality, both for loss and correlations. However, the improved performance of the latest deep learning-based methods compared to alignment quality (identity) was not significantly different (*p* > 0.01) for loss. From the viewpoint of MAE, the performances of classical machine learning-based methods were comparable to those of the latest deep learning-based methods, likely because most of these methods were not designed to output GDT_TS itself. Instead, most methods estimate an average lDDT. The MAE values were not sufficiently small, but they could be used as guidelines to estimate the absolute quality of the structure models.

The results for the multi-domain dataset are listed in Table 2. There was no significant differences in the results for the single domain dataset. However, all MQA methods performed better than alignment quality for loss. Indeed, the latest deep learning-based methods were significantly different from alignment quality (identity) in terms of all metrics.

The detailed results with *p*-values are shown in Appendix A. In addition, the results when RMSD is used as a label are shown in Appendix A.

## 4. Discussion

### 4.1. Differences in the Quality of Models with the Same Template

In this study, we generated multiple predicted structures from a single template structure. It is possible to select a template that is most likely to be similar to the target from multiple template candidates based on the alignment quality between the template and the target sequences. However, the quality of the template alignment cannot be used to rank multiple structures predicted from the same template and alignment. Thus, we first analyzed the extent to which the quality differed among the model structures predicted from the same template. We also examined whether the MQA method can rank model structures predicted from the same template.

The difference between the best and worst template models in the single-domain dataset is shown in Appendix A. Note that we only show the difference in quality between the structures predicted from the template for which the best model was derived. The difference for most of the templates was small (less than 0.025), but some were large, with a maximum difference of approximately 0.175. Therefore, it is important to rank multiple structures predicted from the same template. The results for all templates are shown in Appendix A. The same trend was observed for all templates.

We tested whether the MQA method can select the best model among the model structures generated from the same template. For testing purposes, we used the seven templates in Appendix A where the difference between the best and worst models was greater than 0.05. GDT_TS loss was used as the evaluation metric. To compare with the case of random selection, we calculated and compared the expected value. We also compared the case where the worst model was selected. The results of this test are shown in Figure 4. As shown in Figure 4, most MQA methods tend to select better models than random selection. However, there were some cases where the worst model was selected, and there was no method that could always select a model close to the best model.

### 4.2. Situation-Specific MQA Performance Analysis

The results of the MQA performance comparison show that the current MQA methods perform better overall. However, in some cases, it is better to choose a model based on alignment quality rather than MQA methods. Therefore, we analyzed when the alignment quality was effective in selecting a model and when the MQA method was effective.

First, we created categories based on the distribution of the alignment quality of the templates for each target and then compared the performance for each category. We created the following three categories.

Single top: ${\mathrm{identity}}_{1\mathrm{st}}$—${\mathrm{identity}}_{2\mathrm{nd}}>0.1$Multi top: ${\mathrm{identity}}_{1\mathrm{st}}$—${\mathrm{identity}}_{\left[\mathrm{len}(\mathrm{templates})\;\times \;0.1\;+\;0.5\right]}>0.1$ and not single topNo identical top: ${\mathrm{identity}}_{1\mathrm{st}}$—${\mathrm{identity}}_{\left[\mathrm{len}(\mathrm{templates})\;\times \;0.1\;+\;0.5\right]}\leq 0.1$

Here, ${identity}_{Nth}$ denotes the Nth highest identity, and len(templates) is the number of the template structures for a target. These categories were created based on the assumption that it would be better to select a model based on its identity when there is a template with an outstandingly high identity among the templates and to select a model based on the MQA score when there are multiple templates with high identity. The results for each category in the single-domain dataset are shown in Table 3, and the results for the multi-domain dataset are listed in Appendix A. From the loss of each category, it was found that it is best to select a model based on identity when there is a template with exceptionally high identity. In the other cases, MQA methods were found to better select a structure model. Therefore, although identity is an important indicator that affects the model quality, it does not represent the detailed model quality, so model selection using MQA method is better when the difference in identity is less than approximately 10%.

In addition to the distribution of identity for each target, the following four categories were created based on the maximum identity value, and the performance of each category was compared.

High: 0.8≤MaximumidentityMid-high: 0.6≤Maximumidentity<0.8Mid-low: 0.4≤Maximumidentity<0.6Low: Maximumidentity<0.4

The results for each category in the single-domain and multi-domain datasets are shown in Table 4 and Appendix A, respectively. In the single-domain dataset, identity-based selection was best when the identity was 0.8 or higher. Otherwise, MQA method-based selection was better. In many cases, when sequence identity is quite high (over 0.8), the quality of the predicted structure is high, thus if we select the model with the best identity, the model is quite close to the best model. If not, it is better to use the MQA method because it is difficult to identify which models are highly accurate.

We also compared the performance for each class of targets in the single-domain dataset and for each number of domains of the targets in the multi-domain dataset, but there was no significant difference in performance. Therefore, there is little difference between MQA methods that are superior in estimating the quality of a particular structure (e.g., a structure with many alpha helix). This may be due to the many methods using the same dataset for training. As for the number of domains, it is difficult to discuss the performance difference between the MQA methods because the domain number of most targets was two. Detailed results are shown in Appendix A.

## 5. Conclusions and Future Work

In this study, we constructed two MQA benchmark datasets: a single-domain dataset named HMDM-single and a multi-domain dataset named HMDM-multi. Homology modeling was used as the modeling method, and the datasets contained high-quality models for each target. We compared the performance of the existing MQA methods using the constructed datasets and showed that the latest deep learning-based methods performed better. Thus, it is better to consider the score of the MQA method when selecting one structure from multiple predicted structures. However, the deep learning-based method is not superior for all targets. The sequence identity is a good indicator for selecting the best model when there are templates whose identity is much higher than other templates or when there are templates whose identity is quite high. Therefore, it is better to use the MQA method or sequence identity to select the best predicted structural model depending on the situation. The constructed datasets can be downloaded from http://www.cb.cs.titech.ac.jp/hmdm (accessed on 12 March 2022).

One of the future works is to improve the dataset construction protocol. The current number of targets is acceptable considering the target selection method, but increasing the number of targets in the future will enable more reliable evaluation. The sampling method of the model structure can also be improved. There are no significant problems with the current sampling method, but since it is a simple method, further reducing the bias in accuracy will enable better evaluation. Other improvements could be achieved in the selection of multi-domain targets. In homology modeling, we assume that there is no significant difference in orientation between domains in a family, and we do not assign any specific restrictions to the targets. However, there are cases where the orientation differs, thus it is required to select targets based on the strength of the interaction between domains. Furthermore, the addition of targets with no high-identity templates can be considered as future work. There are few targets in this dataset for which only low identity (<30%) templates exist. The ability to select the best structure for such targets is important in real-life applications, thus we will need to add such targets in the future.

Another possible future work is to verify the accuracy estimation performance for AlphaFold2 structures. Recently, the release of AlphaFold2 has made it possible to predict 3-dimensional structures with high accuracy without using homology modeling. Since it is expected that AlphaFold2 structures will be used more in practical applications, it is important to verify the accuracy estimation performance for AlphaFold2 structures.

## Figures and Tables

**Figure 1 bioengineering-09-00118-f001:**
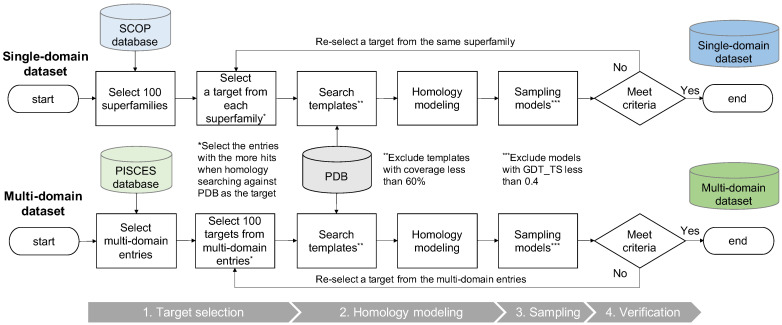
Workflow of dataset creation. First, we selected entries with rich templates from the SCOP and PISCES databases for both single-domain and multi-domain datasets as targets. Next, we performed a template search and homology modeling. In this process, templates with coverage of less than 60% are excluded. The models were sampled for each target so that the distribution of the quality of the models was not biased. When sampling, models with GDT_TS less than 0.4 were excluded. Finally, each target was confirmed to meet the criteria. Targets not meeting the criteria were re-selected.

**Figure 2 bioengineering-09-00118-f002:**
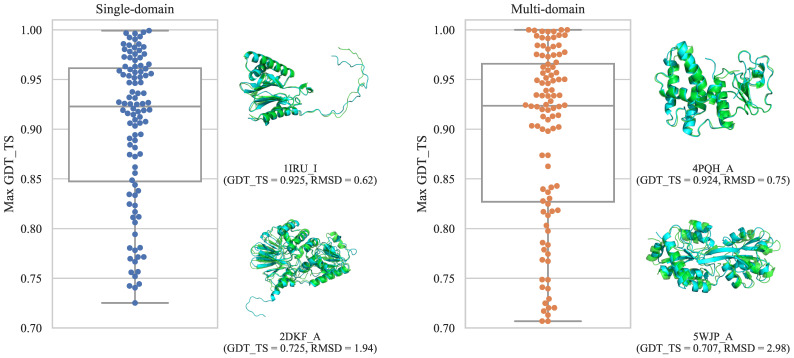
The distribution of the maximum GDT_TS for each dataset. A single point represents the maximum GDT_TS for a single target. For each dataset, superpositions of the native structure and the best structure in the target with the median and the lowest maximum GDT_TS are shown. Native structures are shown in green, and predicted structures are shown in cyan. The superpositions were created using TM-score and structure visualizations were created by PyMOL [48].

**Figure 3 bioengineering-09-00118-f003:**
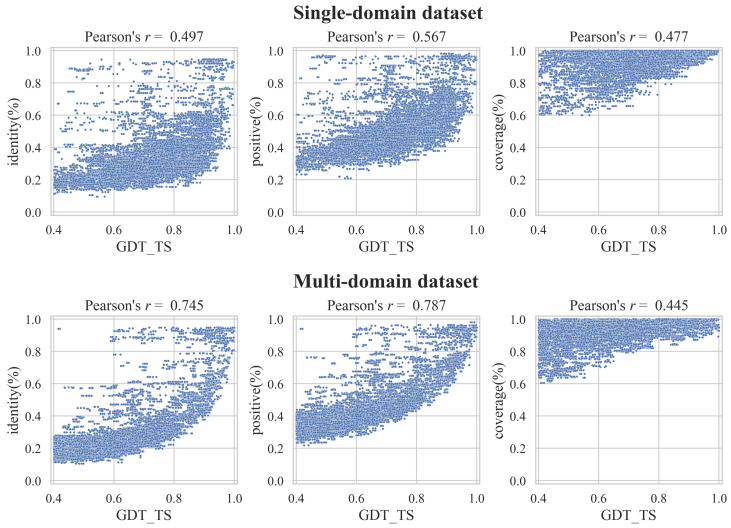
Scatter plot between the model and alignment quality. The first row of the plot is for the single-domain dataset, and the second row is for the multi-domain dataset. Columns 1 to 3 of the plot are identity, positive, and coverage, respectively. The range of GDT_TS in the plot is 0.4 to 1.

**Figure 4 bioengineering-09-00118-f004:**
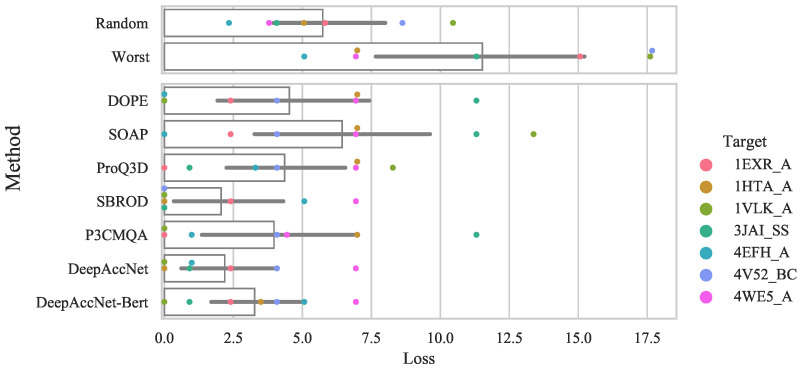
The bar plot and swarm plot of the GDT_TS loss for structure models with GDT_TS difference greater than 0.05 within the same template. The labels on the x-axis represent the method name. In addition to the values of the MQA methods, the values obtained by random selection and the values when the worst model is selected are shown. The bar graph shows the average value of the method, and the error bar represents the 95% confidence interval.

**Table 1 bioengineering-09-00118-t001:** MQA performance for the single-domain dataset.

Method	Loss	MAE	Pearson	Spearman
identity(%)	4.096	(0.371)	0.636	0.507
positive(%)	4.902	(* 0.215)	* 0.661	* 0.540
coverage(%)	* 10.068	(* 0.211)	* 0.438	* 0.359
DOPE	4.013	-	* 0.745	* 0.675
SOAP	3.818	-	0.642	* 0.603
ProQ3D	4.562	* 0.129	* 0.725	* 0.663
SBROD	5.797	-	0.676	* 0.613
P3CMQA	**3.091**	* **0.096**	* **0.838**	* **0.777**
DeepAccNet	3.288	* 0.238	* 0.748	* 0.675
DeepAccNet-Bert	3.372	* 0.173	* 0.821	* 0.754

The first column represents the method name. The second column shows the average GDT_TS loss of the selected models for each target. The values are multiplied by 100 for clarity. The third column shows the average mean absolute error (MAE) between the GDT_TS and estimated scores per target. The fourth and fifth columns show the average Pearson and Spearman correlation coefficients for each target, respectively. The MAE values for identity, positive, and coverage are given in parentheses because they are not scores that directly predict the quality of the model structures. The best values are in bold. An asterisk indicates values for which the *p*-value (calculated by the Wilcoxon signed-rank test against identity) was less than 0.01.

**Table 2 bioengineering-09-00118-t002:** MQA performance for the multi-domain dataset.

Method	Loss	MAE	Pearson	Spearman
identity(%)	4.885	(0.318)	0.787	0.551
positive(%)	4.410	(* 0.171)	* 0.805	* 0.577
coverage(%)	* 16.252	(0.285)	* 0.424	* 0.387
DOPE	* 2.468	-	0.809	* 0.712
SOAP	* 2.921	-	* 0.741	* 0.620
ProQ3D	3.587	* 0.095	0.817	* 0.723
SBROD	3.684	-	0.785	* 0.676
P3CMQA	* **1.884**	* **0.075**	* **0.884**	* **0.802**
DeepAccNet	2.873	* 0.194	* 0.858	* 0.734
DeepAccNet-Bert	* 2.760	* 0.142	* 0.882	* 0.788

The first column represents the method name. The second column shows the average GDT_TS loss of the selected models for each target. The values are multiplied by 100 for clarity. The third column shows the average mean absolute error (MAE) between the GDT_TS and estimated scores per target. The fourth and fifth columns show the average Pearson and Spearman correlation coefficients for each target, respectively. The MAE values for identity, positive, and coverage are given in parentheses because they are not scores that directly predict the quality of the model structures. The best values are in bold. An asterisk indicates values for which the *p*-value (calculated by the Wilcoxon signed-rank test against identity) was less than 0.01.

**Table 3 bioengineering-09-00118-t003:** MQA performance for each category based on the distribution of identity for the single-domain dataset.

Category	Num Targets	Method	Loss	Pearson	Spearman
Single top	9	identity(%)	**1.900**	0.709	0.511
		positive(%)	**1.900**	0.734	0.554
		ProQ3D	4.348	0.877	0.744
		P3CMQA	3.833	**0.926**	**0.821**
		DeepAccNet	3.573	0.855	0.757
		DeepAccNet-Bert	2.539	0.914	0.808
Multi top	41	identity(%)	3.177	0.661	0.481
		positive(%)	3.659	0.693	0.533
		ProQ3D	4.064	0.699	0.660
		P3CMQA	2.459	**0.822**	**0.769**
		DeepAccNet	**2.070**	0.752	0.671
		DeepAccNet-Bert	3.159	0.817	0.752
No identical top	50	identity(%)	5.244	0.602	0.528
		positive(%)	6.461	0.623	0.542
		ProQ3D	5.008	0.718	0.651
		P3CMQA	**3.475**	**0.836**	**0.775**
		DeepAccNet	4.237	0.725	0.664
		DeepAccNet-Bert	3.697	0.807	0.745

The first column represents the name of the category based on the distribution of sequence identity. The second column shows the number of the targets for each category.

**Table 4 bioengineering-09-00118-t004:** MQA performance for each category based on the maximum identity for the single-domain dataset.

Category	Num Targets	Method	Loss	Pearson	Spearman
High	24	identity(%)	**4.357**	0.726	0.581
		positive(%)	4.788	0.751	0.607
		ProQ3D	6.376	0.659	0.605
		P3CMQA	5.377	**0.814**	**0.731**
		DeepAccNet	5.014	0.732	0.636
		DeepAccNet-Bert	5.760	0.762	0.672
Mid-high	23	identity(%)	3.544	0.652	0.498
		positive(%)	4.014	0.682	0.536
		ProQ3D	4.543	0.753	0.703
		P3CMQA	**1.981**	**0.866**	**0.814**
		DeepAccNet	2.336	0.776	0.712
		DeepAccNet-Bert	3.056	0.848	0.781
Mid-low	39	identity(%)	3.969	0.623	0.504
		positive(%)	4.679	0.643	0.541
		ProQ3D	4.413	0.731	0.669
		P3CMQA	2.695	0.825	0.772
		DeepAccNet	2.758	0.731	0.659
		DeepAccNet-Bert	**2.598**	**0.830**	**0.773**
Low	14	identity(%)	4.908	0.491	0.402
		positive(%)	7.175	0.525	0.426
		ProQ3D	**1.896**	0.771	0.681
		P3CMQA	2.096	**0.870**	**0.808**
		DeepAccNet	3.373	0.776	0.725
		DeepAccNet-Bert	1.954	0.851	0.796

The first column represents the name of the category based on the maximum value of identity per target, and the second column shows the number of the targets for each category.

## Data Availability

The datasets can be downloaded at http://www.cb.cs.titech.ac.jp/hmdm (accessed on 9 March 2022). The source code and the dataset are available at https://github.com/yutake27/HMDM (accessed on 9 March 2022).

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
