# Peer review of "A Benchmark Dataset for Evaluating Practical Performance of Model Quality Assessment of Homology Models"

_bioengineering, 2022, doi:10.3390/bioengineering9030118_

Round 1

Reviewer 1 Report

The authors describe a benchmark data set that can be used to evaluate the performance of methods estimating the accuracy of protein homology models. The data set is valuable to the structural biology community and the methods and results are, in general, well presented for the reader familiar with the field.

I have only some minor suggestions that the authors might consider  incorporating into the manuscript, especially to make the description more accessible to the non-specialist reader to increase the impact of the work.

  • I suggest to make the objective of the data set very clear from the beginning. I mean that the accuracy of a homology model can be measured by comparing it to the actual experimental structure and can be estimated without without using the "ground truth" structure. The methods benchmarked here perform the latter task and their performance is evaluated on the basis of the former knowledge. I am aware that this is trivial for the authors but describing this aspect didactically can help the non-specialist reader to get the point of the work, in my opinion.
  • I suggest that the authors discuss the size of their data set (100+100 structures) - do we expect that a much larger data set would be feasible and more helpful in the future, or do the authors justify that the sampling of the model qualities is already very good in their current set? Also, are all the actual experimental structures determined by X-ray crystallography even in the single-domain data set?.
  • Can the authors comment on the usability and performance of GDT_TS on their multidomain data set in the light of potentially different interdomain orientations in the models? 
  • I would welcome a few more sentences, provided it is feasible, on the performance of AlphaFold2's pLDDT on a relevant part the data set.
  • I am fully aware that GDT_TS is a measure superior to RMSD in model evaluation, still, I would like to ask the authors to consider and maybe briefly discuss whether any RMSD-based measure could be used as a comparison. Because energy-based evaluations use side chain-side chain interactions, a heavy atom (not just CA) RMSD might be a meaningful metric that might relate to this. 
  • Finally, I think that the manuscript could benefit from providing visualization of some of the actual domain and model structures and their metrics, to get an impression on the structural differences the given measures can indicate in same sample cases.

Reviewer 2 Report

In this manuscript, the authors create data sets for model quality assessment (MQA) and use them for comparing different MQA methods. They show that deep learning based methods perform better than template sequence identity or classical statistical potentials below the threshold of 0.8 sequence identity.

The model quality assessment problem is of interest for people in the field of molecular modeling but I have concern about the approach both for the construction of data sets and the MQA evaluation.

  1. Construction of data sets
    1. Information on the protocol used for generating models with MODELLER is missing. This should be detailed. In particular, it is mandatory to explain how the five models by alignment are built. Is there a refinement method used? Which one?
    2. For comparing models or methods to assess models, high quality models should be generated. For this purpose, I highly recommend to generate up to 50 or 100 models with refinement “refine.slow” or “refine.fast” and then select the best five models using the pdf score to insure the best models with the provided alignment. The DOPE score is useful for comparing models built from different alignments but the pdf score is very fine for comparing models built from the same alignment after refinement.

  1. MQA evaluation

As pointed by the authors, “it is best to select a model based on identity when there is a template with exceptionally high identity.” I fully agree with this assumption. Concern arises from the categories of sequence identity for comparing MQA methods. The authors consider high identity (>0.8), middle identity (0.6 to 0.8) and low identity (<0.6). This limit of 0.6 is very high for people involved in molecular modeling.

Usually, modeling above a limit of 0.30 for sequence identity is very comfortable. Comparison of MQA methods makes sense only for “true” low identity modeling. The authors should thus compare the performance of the methods for “true” low identity, for example with the limit of 0.30 instead of 0.60.

Reviewer 3 Report

I am a bit concerned about 2 tools that are barely mentioned in your paper.

The first is CAMEO, which you cite, but don't mention their CAMEO-QE: Model Quality Estimation server that runs each week. You explain in detail how CASP doesn't have enough homology modelling targets, but CAMEO-QE has run over 1000 high-quality targets in the past year, where they show that the QMEANDisCo 3 performs best on their Model Quality Estimation tests: https://www.cameo3d.org/quality-estimation/1-year/quality/high

The second tool, that I am concerned that you did not mention at all, is AlphaFold (AF). While your first citation is the AF paper, it is only in reference to your phrase "Various prediction methods have been developed to date", which doesn't mention anything about AF's amazing results in the field. I don't think that you can discuss "using deep learning" without at least a paragraph on AF, especially since they produce an AlphaFold confidence as well.

Minor fixes: I would define what CC stands for in Figure 3, and in Figure S2: I would point out the range of the y-axis in the caption (otherwise, without zooming in it looks like the GDT_TS ranges from 0 to 1), just mention that the GDT_TS ranges from 0.4-1.0 for each plot.

Round 2

Reviewer 2 Report

The authors improved the clarity of the data set objective and the discussion on the limits of the present analysis.

For consistency of the data, the results for the ProQ3D method should be added in Table 3.

MQA methods are very useful when no high identity template is available. In future work, data sets from “truly” low templates (<30%) should be built to compare methods in “real life” conditions. This might be added in the “Conclusion and future work” part.  
